# Metabolic Profiling of *Inga* Species with Antitumor Activity

**DOI:** 10.3390/molecules27154695

**Published:** 2022-07-22

**Authors:** Nerilson M. Lima, Gagan Preet, Sara R. Marqui, Thaís de O. R. S. Falcoski, Geovana Navegante, Christiane P. Soares, Teresinha de Jesus A. S. Andrade, Felipe A. La Porta, Harinantenaina Liva R. Rakotondraie, Marcel Jaspars, Dulce H. S. Silva

**Affiliations:** 1Núcleo de Bioensaios, Biossíntese e Ecofisiologia de Produtos Naturais (NuBBE), Departamento de Química Orgânica, Instituto de Química, Universidade Estadual de São Paulo, Araraquara 14800-900, Brazil; saramarq@iq.unesp.br (S.R.M.); dhsilva1@gmail.com (D.H.S.S.); 2Marine Biodiscovery Centre, Department of Chemistry, University of Aberdeen, Aberdeen AB24 3UE, Scotland, UK; g.preet.18@abdn.ac.uk (G.P.); m.jaspars@abdn.ac.uk (M.J.); 3Faculdade de Ciências Farmacêuticas, Departamento de Análises Clínicas, Universidade Estadual de São Paulo, Araraquara 14801-902, Brazil; thais_tor@yahoo.com.br (T.d.O.R.S.F.); geonavegante@gmail.com (G.N.); soarescp_@fcfar.unesp.br (C.P.S.); 4Núcleo de Investigação Aplicado às Ciências (NIAC), Instituto Federal do Maranhão, Presidente Dutra 65635-468, Brazil; teresinha.andrade@ifma.edu.br; 5Laboratório de Nanotecnologia e Química Computacional (NanoQC), Programa de Pós-Graduação em Ciências de Materiais e Engenharia, Universidade Tecnológica Federal do Paraná, Londrina 86036-370, Brazil; felipe_laporta@yahoo.com.br; 6Division of Medicinal Chemistry and Pharmacognosy, College of Pharmacy, The Ohio State University, Columbus, OH 43210, USA; rakotondraibe.11@osu.edu

**Keywords:** metabolomic, proteomic, bioactive polyphenols, reactive oxygen species, antigenotoxicity, cytotoxicity

## Abstract

This work evaluated the metabolic profiling of *Inga* species with antitumor potential. In addition, we described the antigenotoxicity of polyphenols isolated from *I. laurina* and a proteomic approach using HepG2 cells after treatment with these metabolites. The in vitro cytotoxic activity against HepG2, HT-29 and T98G cancer cell lines was investigated. The assessment of genotoxic damage was carried out through the comet assay. The ethanolic extract from *I. laurina* seeds was subjected to bioassay-guided fractionation and the most active fractions were characterized. One bioactive fraction with high cytotoxicity against HT-29 human colon cancer cells (IC_50_ = 4.0 µg mL^−1^) was found, and it was characterized as a mixture of *p*-hydroxybenzoic acid and 4-vinyl-phenol. The *I. edulis* fruit peel (IC_50_ = 18.6 µg mL^−1^) and *I. laurina* seed (IC_50_ = 15.2 µg mL^−1^) extracts had cytotoxic activity against the cell line T98G, and its chemical composition showed a variety of phenolic acids. The chemical composition of this species indicated a wide variety of aromatic acids, flavonoids, tannins, and carotenoids. The high concentration (ranging from 5% to 30%) of these polyphenols in the bioactive extract may be responsible for the antitumor potential. Regarding the proteomic approach, we detected proteins directly related to the elimination of ROS, DNA repair, expression of tumor proteins, and apoptosis.

## 1. Introduction

Cancer is currently the second most common cause of death globally. In recent years, there has been a significant increase in the number of reported cancer cases [1,2]. According to the International Agency for Research on Cancer (IARC), in 2018 the five-year prevalence of cancer was estimated to be 43.8 million cases, and there were 9.6 million deaths due to this disease [1,2].

Because of the aggressive nature of cancer and limited therapeutic options, together with the toxicity and ineffectiveness of available therapies, the search for new anticancer molecules is essential. Many natural products and their semisynthetic analogues derived from plants have recognized antitumor activity, indicating that searching for new anticancer agents from natural sources may be very promising [3]. Due to easy access to medicinal plants, most discoveries of anticancer compounds have come from plant species and have been of critical importance for the discovery of new cancer therapeutic agents, with the prospect of further discoveries and innovations with the advent of new technologies [4].

Phytochemicals isolated from Fabaceae species are promising candidates for the development of anticancer agents [5]. Among the species with recognized pharmacological activity is the genus *Inga*, whose fruits are widely used as food in the Amazon region; in folk medicine as antipyretic, astringent, antiarrhythmic, antirheumatic agents; and as intestinal washes [5,6,7,8]. The genus *Inga* shows a wealth of polyphenols with remarkable pharmacological properties, including antitumor activity [5].

*Inga* species are widely explored as an important source of bioactive flavonoids, such as myricetin-3-O-α-rhamnopyranoside and myricetin-3-O-(2″-*O*-galloyl)-α-rhamnopyranoside, which have shown potent antioxidant properties with a notable free radical neutralization activity [5,9]. It is well known that the most reactive oxygen species (ROS) are usually formed under normal physiological conditions and, hence, can readily attack DNA, promoting a large variety of lesions [10]. Consequently, these lesions (e.g., oxidized bases and filament breakages) lead to mutagenesis and/or cell death, especially in the case of cytotoxic lesions that block the progression of DNA/RNA polymerases [11].

To isolate bioactive molecules, bioassay-guided fractionation is commonly employed [12]. This has led to the study of a wide variety of bioactive substances present in fractions and subfractions during the isolation process [12]. Hence, this strategy was used in this work to isolate and identify antitumor metabolites from *Inga* species. This study aimed to obtain a metabolic profiling of three *Inga* species with antitumor activity, evaluate the antigenotoxicity of the flavonoids isolated from bioactive *I. laurina* fractions, and apply a proteomic approach to verify which proteins are differentially expressed in cell lines following treatment with these flavonoids.

## 2. Results and Discussion

### 2.1. Bioassay Guided Fractionation

The active extract/fractions/sub-fractions/compound were assessed for cytotoxicity against HepG2, HT-29, and T98G cancer cell lines using doxorubicin as a positive control. The flavonoids isolated from the bioactive fractions from *I. laurina* exhibited promising results of cytotoxicity, antigenotoxicity and chemoprevention, using HepG2 cancer cells [13]. Table 1 and Table 2 show the results of the cytotoxicity assessment against the HT-29 and T98G cancer cell lines. Table 1 shows that the most active extracts against HT-29 were the samples of the branch extracts of *I. marginata* (IC_50_ =13.8 µg mL^−1^) and *I. laurina* seeds (IC_50_ = 15.2 µg mL^−1^). The other extracts presented inhibitory concentration values, IC_50_ > 20.0 µg mL^−1^.

The ethanolic extract of *I. marginata* branches showed promising antitumor activity (IC_50_ = 13.8 µg mL^−1^). Hence, it was selected for the purification of active metabolites. The crude extract was partitioned with *n*-Hexane and MeOH, and the two fractions obtained were evaluated for their antitumor potential. The methanolic fraction showed better activity (IC_50_ = 7.4 µg mL^−1^) when compared with the *n*-Hexane fraction (IC_50_ = 20.0 µg mL^−1^), and it was, therefore, subjected to purification by reversed-phase medium pressure liquid chromatography (MPLC) using a step gradient of 40% MeOH, 70% MeOH, and 100% MeOH. Five fractions were obtained and analyzed by HPLC-PDA to obtain a chemical profile. This analysis allowed the isolation and identification of the compounds *p*-coumaric acid (**1**), gallic acid (**2**), and myricetin-3-*O*-rhamnopyranoside (**3**).

The ethanolic extract of *I. laurina* seeds (IC_50_ = 15.2 µg mL^−1^) presented cytotoxic potential and, therefore, it was submitted to chromatography to isolate the bioactive constituents (Figure 1). The fractions from liquid–liquid partition (*n*-Hexane, EtOAc, and H_2_O) were subjected to analysis by TLC, RP-HPLC-PDA, and NMR. The ^1^H-NMR spectrum of the EtOAc fraction indicated a complexity mixture of low polarity metabolites and evidence of nitrogen compounds. The aqueous fraction presented a variety of phenolic compounds rich in free sugars, and the *n*-Hexane fraction showed a complex mixture of aliphatic substances, such as terpenes and fatty acids. To obtain the bioactive phenolic compounds from the aqueous fraction, this fraction was subjected to chromatographic purification using Diaion HP. The fraction obtained in pure MeOH (fraction 3) showed 1-NMR signal related to alkaloids that was not yet identified in the *Inga* genus. However, it did not show cytotoxic potential in the assay that was performed. Fractions 1 (40% MeOH) and 2 (70% MeOH) showed IC_50_ of 9.2 µg mL^−1^ and 7.6 µg mL^−1^, respectively. The ^1^H-NMR spectra were obtained for these three fractions to observe the common signals and to identify the substances responsible for cytotoxic potential. A comparative analysis of the signals present in the spectrum of the most active fraction (fraction 2) with the non-active fraction (fraction 3) showed that the low concentration constituents (information that was obtained by integrating the signals) are responsible for the cytotoxic potential. In addition, the ^1^H-NMR spectrum of fraction 2 showed signs of at least two aromatic substances. The active fraction (800 mg) was subjected to column chromatography. The most polar fraction obtained in 100% MeOH showed excellent cytotoxic activity (IC_50_ = 4.0 µg mL^−1^) in the evaluated assay. ^1^H-NMR spectroscopic analysis of this fraction showed a mixture of two phenolic compounds, *p*-hydroxybenzoic acid (**4**) and 4-vinyl phenol (**5**), which was mainly identified by characteristic signs of the vinyl unit and by aromatic protons. Its methoxylated derivative, 4-vinyl-2-methoxy-phenol, has been widely reported in Fabaceae species [14]. To the best of our knowledge, this is the first report of these compounds in the genus *Inga*. Fraction 1 (240 mg) was subjected to Sephadex LH-20 exclusion chromatography and its fractions were submitted to TLC and HPLC-PDA analysis. The analysis of the UV spectra of the most resolved peaks of fractions 2 and 5 suggested the presence of organic acids (bands at 270nm to 280 nm) and catechins (bands around 285 nm), respectively. Compounds that have a catechol skeleton in their structure (natural or synthetic) may have potential in the treatment of gliomas [15].

The extracts of plant material from *Inga* species were also submitted to cytotoxicity analysis against T98G human glioblastoma cancer cell lines, using doxorubicin as the positive control (IC_50_ = 2.10 µg mL^−1^). The T98G human glioblastoma cancer cell lines were used to verify the effectiveness of the cytotoxic potential from *Inga* species in other tissue types of aggressive cancers. Table 2 shows that except for the extract of *I. laurina* seeds and *I. edulis* fruit peel, the plant material obtained from the three species of *Inga* showed no cytotoxic effect against this tumor cell line. *I. laurina* seed extract showed antitumor potential in the assay performed against HT-29 human colon cancer lines and was submitted to other chromatographic procedures to characterize their small molecules. From the study of the *I. laurina* seeds, a mixture of two aromatic substances with a significant cytotoxic effect was obtained (**4**, **5**). However, further studies are needed to clarify which substances are responsible for this effect. The following compounds were identified in this extract: methyl gallate (**6**), myricetin-3-O-rhamnopyranoside (**3**), and myricetin-3-O-(2″-*O*-galloyl)-α-rhamnopyranoside (**7**).

The extract of *I. edulis* fruit peel was submitted to liquid–liquid partition, providing *n*-Hexane and MeOH fractions. These fractions were evaluated in the same bioassay, and only the MeOH fraction (IC_50_ = 12.0 µg mL^−1^) presented cytotoxic potential. Thus, it can be inferred that the substances responsible for the antitumor potential have higher polarity. To obtain a chemical profile and to verify the chemical composition of the most active material, these fractions were analyzed by HPLC-PDA under the same conditions previously employed. Analysis of the most active chromatographic fraction (MeOH) showed a high content of phenolic compounds, especially flavonoids and organic acids. A more detailed analysis of retention times and UV spectra of the most resolved peaks, compared with previously isolated substances from other species, allowed us to verify the presence of benzoic acid (Rt = 24.6 min; λ: 221/273 nm) (**8**) and vanillic acid (Rt = 18.45 min; λ: 261/291 nm) (**9**). In the lowest polarity fraction, the presence of other compounds that were not previously identified in the species was observed. The UV spectrum of these compounds obtained by RP-HPLC-PDA analysis under the same conditions used previously showed the presence of five pigments. The peaks related to these substances showed, in addition to the band at 225 nm, another band at around 400 nm. These substances are possibly pigments that occur in these tissues, such as carotenoid derivatives that are already characterized in *Inga* species [16]. The carotenoids absorb light in the region of 400nm to 500 nm of the visible spectrum, whose characteristic provides the yellow/red colors to these pigments [17]. The metabolites identified in the cytotoxic *Inga* species are shown in Figure 2.

Several studies have reported natural phenolic compounds as potent bioactives with potential for cancer prevention [18,19,20]. The phenolic compounds identified in the bioactive fractions from *Inga* species are reported in the literature with cytotoxic activity against several tumor cell lines, such as Gallic acid [21], *p*-hydroxybenzoic acid [22], vanillic acid [23], *p*-coumaric acid [24], and benzoic acid [25]. Although its methoxylated derivative 2-methoxy-4-vinylphenol is experimentally established to have cytotoxic activity [26,27], this is the first report of the cytotoxic activity of 4-vinyl phenol. Therefore, further studies are needed to confirm its bioactivity. Regarding the biologically active flavonoids (myricetin-3-O-α-rhamnopyranoside and myricetin-3-O-(2″-*O*-galloyl)-α-rhamnopyranoside) described in this study, their structure–activity relationship can be made on the basis of the important role of the C_2_C_3_ double bond that contributes to molecular planarity and conjugation between rings C and A/B, which leads to the potent biological activity of these compounds, as shown in previous studies. comprising potent tumor inhibition in flavonoids [28,29]. The structures of these two flavonoids exhibit the co-existence of C_2_C_3_ unsaturation and two ring B hydroxyl groups that lead to a greater inhibition effect, as shown in a previous study [29]. In this study, flavonoid structures have a contributive role of 5,7-diOH on ring A hydroxylation, which contributes to stronger inhibitory activity on cancer cells. Many reports have provided evidence about the influence of hydroxylation on tumor modulation [30]. In addition, the presence of ring B substitution, such as catechol moiety, plays a vital role in influencing the biological activity, and similar substitution has been reported in a study [28]. In addition to this ring C, 3-substitution has been considered as important for improving biological effects, as shown in a previous study [31,32].

### 2.2. Quantification of Phenolic Compounds Isolated from Cytotoxic Extracts

An analytical method using high-performance liquid chromatography equipped with a photodiode array detector (HPLC-PDA) was developed and validated for the determination of major phenolic compounds (gallic acid (**2**), methyl gallate (**6**), myricetin-3-O-rhamnoside (**3**), and myricetin-3-O-(2″-O-galloyl)-α-rhamnopyranoside (**7**)) in the cytotoxic extract from *Inga laurina*. The quantification of these metabolites showed an analytical curve with good linearity in the concentration range used, where the values of the correlation coefficient (R)^2^ that were obtained were higher than 0.991; this is in accordance with the minimum criterion acceptable by the regulatory agency (ANVISA), whereby values must be greater than 0.990 (Table 3). The sensitivities of the method obtained were 9.6 × 10^5^, 1 × 10^5^, 5.2 × 10^6^, and 3.4 × 10^6^ for gallic acid, methyl gallate, myricetin-3-O-rhamnoside, and myricetin-3-O-(2″-O-galloyl)-α-rhamnopyranoside, respectively. These sensitivities correspond to the slope of the line (slope of the curve) [33]. The results showed 0.035 mg/mL of gallic acid for each 0.7 mg/mL of EtOAc fraction (equivalent to 5%), 0.131 mg/mL of methyl gallate (equivalent to 18.7%), 0.171 mg/mL of myricetin-3-O-rhamnoside (equivalent to 24.3%), and 0.21 mg/mL of myricetin-3-O-(2″-O-galloyl)-α-rhamnopyranoside for each 0.7 mg/mL of EtOAc fraction (equivalent to 30%).

### 2.3. Antigenotoxicity and Proteomic Approach from the Flavonoids

An antigenotoxicity assessment and a proteomic approach were performed with the bioactive flavonoids myricetin-3-O-(2″-*O*-galloyl)-α-rhamnopyranoside and myricetin-3-O-α-rhamnopyranoside isolated from *I. laurina* leaves and identified by NMR spectroscopy [34].

Concerning the antigenotoxicity evaluation after damage with hydrogen peroxide, we noted that the myricetin-3-O-α-rhamnopyranoside presented a significant difference at concentrations of 1.5 μg/mL, 4.4 μg/mL, and 13.3 μg/mL, respectively: 18.48 ± 27.61, 10.84 ± 14.43, and 12.23 ± 17.9 (% DNA ± SE), respectively, with a significance value lower than 0.001 at the three concentrations evaluated. Myricetin-3-O-(2″-*O*-galloyl)-α-rhamnopyranoside showed significant differences only at concentrations of 1.5 μg/mL and 13.3 μg/mL, 15.77 ± 16.34, and 22.25 ± 29.72, respectively, with a significance value of less than 0.01 for the concentration of 13.3 μg/mL and less than 0.05 for the concentration of 1.5 μg/mL. The % DNA ± SE of the negative control was 10.71 ± 1.05 and the positive control was 36.21 ± 3.33. Previous reports showed that both flavonoids are not capable of producing genotoxic activity and have a protective effect against H_2_O_2_ damage [13].

Based on the antigenotoxicity experiment, the concentrations tested for the proteomics assay were 4.4 µg/mL for myricetin-3-O-α-rhamnopyranoside and 13.3 µg/mL for myricetin-3-O-(2″-*O*-galloyl)-α-rhamnopyranoside. Although the only structural difference between the substances was the presence of the galloyl group, there was a difference in the protein expression profile. Thirty-three differentially expressed proteins with a score ≥ 30 were identified for the two flavonoids evaluated. The proteins found were separated by treatment and grouped based on the increase or decrease in the ratio between treatment and control, where values ≥ 1.5 indicated that treatment increased protein expression, compared with control, and values ≤ 0.5 indicated that the treatment decreased protein expression. Values equal to 1 represented proteins with the same expression in both the control and the treatment groups.

Treatment with the flavonoid myricetin-3-rhamnoside showed differential expression in 18 proteins, with only four proteins showing an increased expression and 14 showing a decreased expression (Table 4). Fifteen differentially expressed proteins were selected following treatment with the flavonoid myricetin-3-O-(2″-*O*-galloyl)-α rhamnopyranoside, of which three showed increased expression and 12 showed decreased expression (Table 4). By grouping the proteins identified in the three treatments, it was observed that of the 31 proteins identified, only nine were directly related to oxidative stress or to the DNA damage response.

Proteins associated with genotoxic agents belong to the DNA repair class and possess other related functions. In our study, the protein ERO1A was regulated by both treatments; its expression was decreased by the compound myricetin-3-rhamnoside and increased by myricetin-3-O-(2″-*O*-galloyl)-α-rhamnopyranoside. This protein exhibits disulfide oxidoreductase activity, catalyzing the formation and rearrangement of the disulfide bonds, with the concomitant reduction of molecular oxygen to form hydrogen peroxide [35]. Two other proteins that are related to the response to oxidative stress are the glial fibrillary acidic protein (GFAP) and vimentin (VIME). The expression of these two proteins was found to be increased following treatment with the myricetin-3-rhamnoside compounds. Recent studies showed that the elimination of ROS is impaired in the absence of these two proteins [36]. The protein NQO1 is an enzyme involved in a number of cellular processes relevant to cancer. It also stabilizes stress response proteins, such as p53 and p73, and modulates the NF-κB pathway [37]. In this study, NQO1 expression was found to be decreased after treatment with the flavonoid myricetin-3-rhamnoside.

Both compounds caused a decreased expression of the protein H4. Liang et al. (2012) [38] showed that an increase in histone gene dosage resulted in an increased sensitivity to DNA damage, whereas the elimination of a pair of H3–H4 genes resulted in reduced levels of free H3 and H4, concurrent with resistance to the DNA-damaging agents. The protein TERA, which was decreased by myricetin-3-rhamnoside, is also related to DNA repair. Therefore, a mutation in the protein gene may decrease its functionality by increasing the amount of unrepaired double-stranded DNA [39]. In contrast, the NPM protein reduction may potentially inhibit DNA repair, considering its use as a therapeutic target [40]. It was found that NPM expression was increased by myricetin-3 rhamnoside, which suggested an increase in the stability of DNA repair.

## 3. Materials and Methods

### 3.1. Plant Material

The species *Inga laurina* (Sw.) Willd, *Inga edulis* Mart., and *Inga marginata* Willd were collected in Assis city (São Paulo State, Brazil) in April 2008 and identified by Dr. Giselda Durigan. A voucher specimen of *Inga laurina* (FEA 3552), *Inga marginata* (FEA3304), and *Inga edulis* (FEA3306) was deposited in the botanical collection of Assis State Forest.

### 3.2. General Experimental Procedure

NMR spectral data were obtained on a Varian Inova-500 instrument, at 125 MHz for ^13^C and 500 MHz for ^1^H. RP-HPLC-PDA analyses were performed on a Shimadzu^®^ chromatograph (Shimadzu SPD-M20A, Kyoto, Japan). Silica gel 60 (230–400 mesh, Merck^®^ (Darmstadt, Germany) was used for chromatographic column and solvents used in the preparation of extracts and fractions were all analytical-grade.

### 3.3. Extraction Procedures

The plant material (leaves, branches, and fruits) from *Inga* species (*Inga laurina*, *Inga marginata*, and *Inga edulis*) were dried at room temperature, ground, and extracted with ethanol for 20 min, using an ultrasonic bath. After evaporation under reduced pressure using a rotary evaporator (54-Rotavapor R-220, brand: Büchi, with vacuum pump Vacuum Controller V-805, brand: Büchi and water circulator), the EtOH extracts were submitted to cytotoxicity evaluation against HT-29 human colon cancer cells (ATCC^®^ HB-8248™, Manassas, VA, USA) and T98G human glioblastoma cancer cell lines (ATCC^®^ CRL-1690™), using doxorubicin as a positive control.

### 3.4. HPLC-PDA Analysis of the Extracts and Fractions Bioactives

All the extracts and their fractions with cytotoxic potential were analyzed by HPLC-PDA (high-performance liquid chromatography coupled with a photo diode array detector) (Shimadzu Chromatograph equipped with two Shimadzu LC-10AD pumps, Shimadzu SIL 10A auto-injector, UV-Vis array detector model Shimadzu SPD/MX/AVP; the data acquisition and processing were treated on the Shimadzu CLASSLC10 software version 1.64A) in analytical mode (gradient elution 5% to 100% MeOH; analytical column “Luna” Phenomenex^®^ C18 (250 × 4.6 mm, 4 μm) and UV detector monitored at 254 nm; injection volume: 40 μL, run time: 40 min, flow: 1 mL min^−1^), in order to rapidly obtain a profiling of the bioactive compounds.

### 3.5. Purification of the Ethanolic Extracts of Inga Species

The ethanolic extracts of the seeds from *I. laurina*, and *I. marginata* branches presented cytotoxic potential against HT-29 human colon cancer cells and, therefore, they were selected for bioguided chromatographic fractionation and chemical investigation for their bioactive components.

The ethanolic extract (130 mg) from *I. marginata* branches was partitioned with *n*-Hexane and MeOH, and the two fractions obtained were evaluated for their cytotoxic potential. The MeOH fraction (100 mg) presented better activity, three times higher than that of the *n*-Hexane fraction (25 mg) and, therefore, it was submitted to purification by medium pressure liquid chromatography (MPLC) in a C18 RP-silica column using elution systems of 40% MeOH, 70% MeOH, and 100% MeOH. Five subfractions (100 mL) were obtained and analyzed by HPLC-PDA to obtain a chemical profile. This analysis allowed for the isolation and identification of *p*-coumaric acid (compound **1**; 3 mg), gallic acid (compound **2**; 27 mg), and myricetin-3-O-rhamnopyranoside (compound **3**; 18 mg).

The crude extract obtained from *I. laurina* seeds (4.5 g) was submitted to liquid–liquid partition with EtOAc and H_2_O, providing 500 mg of organic fraction and 4.0 g of the aqueous fraction. After ^1^H-NMR analysis, the aqueous fraction was subjected to the chromatographic column using Diaion HP-20. Chromatographic fractionation was performed by eluting with water to obtain the free sugars and, then, with MeOH to obtain the other polar compounds. The methanolic fraction (350 mg) was subjected to chromatographic analysis by column chromatography on C18 silica and eluted with the systems of 40% MeOH (fraction 1), 70% MeOH (fraction 2), and 100% MeOH (fraction 3). The three fractions were submitted to bioassay for cytotoxicity evaluation and analyzed by ^1^H-NMR and HPLC-PDA. The fraction obtained in 100% MeOH (fraction 3) did not show cytotoxic potential in the assay, while fractions 1 (MeOH 40%) and 2 (MeOH 70%) presented IC_50_ lower than 10 µg mL^−1^. ^1^H-NMR spectra were obtained from the three fractions in order to observe the common signals and to determine the substances present that were responsible for the cytotoxic potential. Fraction 2 (45 mg) was subjected to column chromatography on silica gel 60 and eluted with EtOAc/MeOH 1:1 to 100% MeOH, yielding four subfraction groups. The polar fraction, obtained in 100% MeOH, showed excellent cytotoxic potential against this tumor cell line. Analysis of the NMR spectra of this fraction showed the presence of only two phenolic compounds, which were identified as a mixture (10 mg) of the compounds *p*-hydroxybenzoic acid (4) and 4-vinyl phenol (5). Fraction 1 (240 mg) was subjected to Sephadex LH-20 exclusion chromatography and eluted with 100% MeOH, providing 10 subfractions. The fractions (2, 3, 4, and 5) that presented better separation in the reverse phase TLC analysis were submitted to HPLC-PDA analysis to obtain a chemical profile to identify the bioactive components. This analysis showed the presence of organic acids and catechins.

All of the extracts of the *Inga* species were subjected to cytotoxicity analysis against another tumor line (human glioblastoma T98G), using doxorubicin as a positive control. Only the seed extracts from *I. laurina* and the extract from the fruit peel of *I. edulis* presented cytotoxic potential against this tumor line and, therefore, only those extracts were fractionated to isolate the components responsible for this biological potential.

The ethanolic extract of *I. laurina* seeds was previously analyzed, where it was possible to identify compounds **4** and **5**. In this extract, the following compounds were identified: methyl gallate (compound **6**; 17 mg), myricetin-3-O-rhamnopyranoside (compound **3**; 11 mg), and myricetin-3-O-(2″-O-galloyl)-α-rhamnopyranoside (compound **7**; 13 mg). The extract from fruit peel (900 mg) of *I. edulis*, which is active against this tumor cell line, was submitted to a liquid–liquid partition, providing two fractions: *n*-Hexane (270 mg) and MeOH (470 mg). The fractions obtained were evaluated under the same conditions previously employed, and the chemical profile obtained indicated the presence of compounds benzoic acid (8; 8 mg), and vanillic acid (9; 7 mg).

### 3.6. Description of the Isolated Compounds

Bioactive compounds were identified by ^1^H-NMR analysis (Varian Inova-500 instrument), HPLC-PDA analysis through retention time and UV spectra with authentic standards Sigma–Aldrich (Jurubatuba, Brazil), and literature data such as chemosystematic studies of natural compounds isolated from *Inga* genus. *p*-coumaric acid (1): ^1^H-NMR (500 MHz, DMSO-*d_6_*) δ: H 7.60 (d, J = 16 Hz, H-8), 6.31 (d, J = 16 Hz, H-7), 7.43 (d, J = 8.4 Hz, H-2/H-6), 6.81 ppm (d, J = 8.4 Hz, H-3/H-5); Gallic acid (2): ^1^H-NMR (500 MHz, DMSO-*d_6_*) δ: 6.92 (s, 2H, H-2/H-6); Myricetin-3-O-rhamnoside (3): ^1^H-NMR (500 MHz, DMSO-*d_6_*) δ: 12.67 (s; OH), 6.89 (s, 2H, H-2′/H-6′), 6.37 (d, J = 2 Hz, H-8), 6.20 (d, J = 2 Hz, H-6), 5.2 (d, J = 1.5 Hz, H-1”), 4.0 (m, H-2″), 3.5 (m), 3.3 (m), 3.1 (m), 0.8 (d, J = 6.5, -CH_3_); *p*-hydroxybenzoic acid (4): ^1^H-NMR (500 MHz, DMSO-*d_6_*) δ: 8.00 (d, J = 8.4 Hz), 6.90 (d, J = 8.4 Hz); 4-vinyl phenol (5): ^1^H-NMR (500 MHz, DMSO-*d_6_*) δ: 7.59 (dd, J = 7.6 and 1.6 Hz), 7.49 (dd, J = 7.6 and 1.6 Hz), 6.97 (d, J = 7.6), 6.93 (d, J = 7.6), 5.94 (dd, J = 17.2 and 10.8 Hz), 5.25 (dd, J = 17.2 and 1.2 Hz), 5.08 (dd, J = 10.8 and 1.2 Hz); methyl gallate (6): ^1^H-NMR (500 MHz, DMSO-*d_6_*) δ: 6.93 (s, 2H, H-2/H-6), 3.73 (s, 3H); myricetin-3-O-(2″-O-galloyl)-α-rhamnopyranoside (7): ^1^H-NMR (500 MHz, DMSO-*d_6_*) δ: 12.60 (s), 6.95 (s, 2H, H-2′/H-6′), 6.92 (s, 2H, H-2′″/H-6′″), 6.38 (s, H-8), 6.20 (s, H-6), 5.50 (d, J = 1.5 Hz, H-1”), 5.47 (dd, J = 1.5 Hz, H-2″), 3.8–3.3 (m), 0.94 (d, J = 5.5 Hz, -CH_3_); benzoic acid (8): ^1^H-NMR (500 MHz, DMSO-*d_6_*) δ: 8.02 (dd; J = 8.4 and 1.5 Hz; 2H), 7.57 (tt; J = 7.5 and 1.5 Hz, 1H), 7.46 (tq; J = 7.5 and 1.5 Hz; 2H); vanillic acid (9): (tr = 18.45 min; λ: 261/291 nm).

### 3.7. Biological Assays

#### 3.7.1. Cytotoxicity Assay

The cells were cultured in DMEM culture medium (Modified Eagle Medium Dulbecco’s) supplemented with penicillin 100 U mL^−1^ and streptomycin 0.1 mg mL^−1^ sodium bicarbonate, kanamycin, and pH 7.2, and supplemented with 10% fetal bovine serum. Briefly, the cells were cultured in bottles maintained at 5% CO_2_ and a temperature of 37 °C until the cell monolayers were confluent at about 70% to 80%.

For the cytotoxicity assay using Sulforhodamine B (SRB), a suspension of the HepG2 cell line containing 1.5 × 10^4^ cells/well was used. Cells were cultured in 96-well plates and after 24 h of cultivation, the pure compounds were added, following a serial 1:3 dilution starting at a concentration of about 200 µg mL^−1^.

After 24 h of treatment, 50 µL of 50% trichloroacetic acid (TCA) was added at a low temperature, and the plates were incubated for 1 h at 4 °C; then, the TCA solution was removed, and the plates were washed with tap water 3 to 4 times. It was added to 50 µL of SRB solution at 0.4% (dilute acetic acid), and the plates were then incubated for 20 min at room temperature. After removal of the SRB, the plates were washed 3 to 4 times with 1% acetic acid, dried, and dissolved in dye with 10 mM Tris Base (Sigma^®^). After 5 min of incubation at room temperature, the spectrophotometric reading of absorbance was performed at a wavelength of 570 nm in the plate reader iMark Microplate Reader (Bio-Rad Laboratories^®^, Hercules, CA, USA).

Tests were performed in three independent experiments, and the percentage of living cells was calculated in relation to the negative control, representing the cytotoxicity of each treatment, as proposed by Zhang et al. in 2004. The same assay cytotoxicity was performed for the HT-29 and T98G cancer cell lines. In this work, we evaluated different cancer cell lines because they are types of cancers that are aggressive, with worse prognosis, and in order to verify whether the bioactive compounds could be used in the future as antineoplastic agents for different cancers and to verify their effectiveness in other types of tissues.

#### 3.7.2. Antigenotoxicity Assessment

The human hepatocellular carcinoma (HepG2; ATCC HB-8065) was kindly provided by Dr. Dayse Maria Favero Salvadori (Department of Pathology of the Faculty of Medicine, UNESP, Botucatu Campus, Botucatu, Brazil). The strain of HepG2 cells was cultured in DMEM culture medium (Modified Eagle Medium Dulbecco’s, Sigma^®^) supplemented with penicillin 100 U/mL and streptomycin 0.1 mg/mL sodium bicarbonate, kanamycin (Sigma^®^), and pH 7.2, supplemented with 10% fetal bovine serum (Cultilab^®^). The cells were cultured in bottles maintained at 5% CO_2_ and a temperature of 37 °C until the cell monolayers were confluent at 70% to 80%.

In order to evaluate the antigenotoxicity of the flavonoids from *I. laurina*, hydrogen peroxide (H_2_O_2_) was used as a positive control at a concentration of 0.01 M in the HepG2 cells. Pretreatment and posttreatment protocols were used. In pretreatment, the cells were treated with the flavonoids and subsequently subjected to H_2_O_2_ damage, while in posttreatment the cells were subjected to damage with H_2_O_2_, and subsequently the flavonoids were added. These two treatment protocols allowed the identification of the antigenotoxic action at different concentrations of the flavonoids that were proposed for this study. For the antigenotoxicity assay, the HepG2 cells were treated with non-cytotoxic concentrations. In the pretreatment, the cell suspension of 2.5 × 10^5^ cells/well was grown in 24 well plates for 24 h with a final volume of 500 μL/well, and then the cells were treated with compounds for 24 h. As a negative control, only a culture medium was used, and as a positive control, the hydrogen peroxide (H_2_O_2_) at 0.01 M for 5 min was used. After 24 h of incubation with the treatments, 0.01 M hydrogen peroxide (H_2_O_2_) was added for 5 min. After the 5 min period, the culture medium was removed and the cells were washed, trypsinized, stored in 1.5 mL tubes, and centrifuged at 1500 rpm for 3 min. After centrifugation, the supernatant was removed and the cells were resuspended in 200 μL of agarose (low melting point) at 37 °C. Then, the volume of each Eppendorf tube was transferred to two slides that were previously treated with normal-melting-point agarose. Each blade was covered with a large coverslip (24 × 60 mm) and placed in the refrigerator under light for 5 min. Then, the coverslips were removed, and the slides were immersed in a freshly prepared lysis solution at 4 °C for a minimum of 12 h under refrigeration. The slides were removed from the lysis solution and subjected to alkaline electrophoresis for 20 min. After that, the slides were neutralized and fixed. The slides were analyzed after staining in a fluorescence microscope. 100-nucleoid images were captured and analyzed by TriTek CometScore TM software version 1.5. The parameter adopted in the present study was % DNA in the Tail [41,42]. In the posttreatment, the cell suspension of 2.5 × 10^5^ cells/well was grown in 24-well plates for 24 h and at a final volume of 500 μL/well; hydrogen peroxide (H_2_O_2_) to 0.01 M for 5 min was added to induce the formation of cellular damage. After this period, the cells were treated with flavonoids for 24 h [42].

#### 3.7.3. Proteomic Analysis

To determine the protein profile, the HepG2 cancer cells, treated and untreated with the flavonoids myricetin-3-O-rhamnopyranoside and myricetin-3-O-(2″-O-galloyl)-α-rhamnopyranoside isolated from *I. laurina*, were used. The flavonoids myricetin-3-*O*-rhamnopyranoside and myricetin-3-O-(2″-*O*-galloyl)-α-rhamnopyranoside were selected for the proteomics analysis in the HepG2 cells, because they are the major compounds of the bioactive fractions and they showed more significant results in assays of chemoprevention, genotoxicity, and cytotoxicity [13]. The proteins were extracted with the following lysis solution: urea 7.7 M, thiourea 2.2 M, CHAPS 4.4%, protease inhibitors (cocktail of inhibitors, 1 µL for 1 × 10^6^ cells, PMSF 1 mM), and phosphatase inhibitors (1 mM sodium orthovanadate, 1 mM NaF, 1 mM sodium β-glycerophosphate, and 10 mM sodium pyrophosphate). For each of the 1 × 10^7^ cells, 100 µL of lysis solution was added. The cells were disrupted for three cycles of (a) ultrasound bath for 5 min and (b) ice bath and vortexing for 5 min; after the third cycle, the samples were frozen and thawed in liquid nitrogen. Then, the total extract was centrifuged at 20,000× *g* for 30 min to precipitate cell debris, and the supernatant was subjected to the determination of the amount of proteins and stored at −80 °C. The quantification of proteins in solution was performed by the method of Bradford (1976) in a microplate. The samples, diluted appropriately in Milli-Q water, were pipetted into a final volume of 20 µL. Then, 200 µL of Bradford reagent (Coomassie Blue G 0.05% in ethanol 25% *v*/*v* and phosphoric acid 50% *v*/*v*) were added. After 15 min of incubation at room temperature, the plate was read at 595 nm in an ELISA reader. Quantification was performed using a standard bovine serum albumin (BSA) curve as a reference [43].

Spectrometric analyses were performed using a MALDI-ToF-ToF mass spectrometer (Axima Performance, Kratos-Shimadzu, Manchester, UK), with the automatic acquisition of the MS and MS/MS spectra for the most abundant ions. Protein identification was performed using the MASCOT program (www.matrixscience.com (accessed on 23 July 2013)). The spectra obtained were processed and submitted to a database for protein identification. For this process, we used the program Mascot version 2.2.04, SwissProt database, taxonomy *Homo sapiens* (human) with the following parameters: enzyme trypsin with loss of one cleavage (“missed cleavage”), fixed modification for Methionil-cys, and variable modifications for methionine oxidation. iTRAQ isobars were also added as variable modifications. To exclude false positive identifications (FDR), the mass spectra were submitted to the database in the reverse mode with a, level of statistical significance of p35, which corresponded to consecutive y or b ions for the sequencing of peptide amino acids by CID-MS/MS [43].

### 3.8. Quantification of Phenolic Compounds from Cytotoxic Extracts

Ultrapure water was obtained from the Mili-Q^®^ purification system. Silica gel adsorbent 60 (230 mesh to 400 mesh) for a chromatographic column was obtained from Merck KGaA, Germany. The solvents used in the preparation of EEIS and the fractions were all analytical grades obtained from Vetec, thin layer chromatography (TLC), silica gel 60 chromate plates with UV254 fluorescence indicator, 0.20 mm thick (MACHEREY-NAGEL-MN). A rotary evaporator (54-Rotavapor R-220, brand: Büchi, with vacuum pump Vacuum Controller V-805, brand: Büchi and water circulator) was used to concentrate the samples. NMR spectral data were obtained on a Varian Inova-500 instrument, at 125 MHz for 13C and 500 MHz for 1 H. silica gel 60 (230 mesh to 400 mesh, Merck^®^ KGaA, Darmstadt, Germany).

RP-HPLC-PDA analysis was carried out in a Shimadzu Chromatograph equipped with two Shimadzu LC-10AD pumps: Shimadzu SIL 10A auto-injector, UV-Vis array detector model Shimadzu SPD/MX/AVP. Briefly, the data acquisition and processing were treated on Shimadzu CLASSLC10 software (version 1.64A). For these experimental measurements at room temperature, we used a Phenomenex C18-Hydro (250 × 4.6 mm, 4 μm) with a flow rate of 1 mL·min^−1^; the injection volume was 30 μL.

In view of the cytotoxicity results obtained, it was considered important to quantify the major compounds that were present in the bioactive extract. Through a qualitative analysis by HPLC-PDA, it was possible to infer the presence of phenolic compounds as being major in the cytotoxic extract; therefore, we proceeded with the quantification of these metabolites isolated from the fraction with the greatest cytotoxic potential (EtOAc). The EtOAc fraction weighed 7 × 10^−1^ mg in an analytical balance. It was dissolved in 1.0 mL of H_2_O:MeOH 55:45 and filtered through a solvent filtration membrane (nylon, 47 mm in diameter, 0.45 Pm, brand: Sigma–Aldrich) and a vial with screw cap and septum (2.0 mL). The materials for quantification were a bottle (vial) with a screw cap and septum (2 mL capacity) for an automatic injector, a volumetric flask (capacity 5 mL), a micropipette (capacity 100, 200 and 1000 PL), a membrane for filtering samples and standards (Millex HV, 0.45 µm pore; 13 mm diameter), brand: Millipore, scale with a sensitivity of 0.001 mg).

The standards gallic acid (compound **2**), methyl gallate (compound **6**), myricetin-3-O-rhamnopyranoside (compound **3**), and myricetin-3-O-(2″-*O*-galloyl)-α-rhamnopyranoside (compound **7**) were prepared by dilution from a stock solution. In this way, the external standard method was used. One mg of each standard substance was weighed on an analytical balance and dissolved in 1.0 mL of H_2_O:MeOH 55:45. A membrane was used for filtration and a vial with a screw cap and 2.0 mL septum. Through these solutions, it was prepared to stock solutions. Starting from the stock solutions, six successive dilutions were made, named working solutions with the aid of 10 mL and 5 mL volumetric flasks. The EtOAc fraction was subjected to HPLC-PDA in analytical mode monitoring at 254 nm, using the “Luna” Phenomenex C-18 analytical column (25 cm × 4.6 mmdi × 5 μm) eluted with different gradients and an isocratic system for observation of the chromatographic profile. The best chromatographic resolution was obtained with a gradient system H_2_O/MeOH 95:5 to H_2_O/MeOH 55:45 (20 min), followed by an isocratic system H_2_O/MeOH 55:45 for 30 min, with a flow of 1 mL/min. Thus, the chromatographic condition for quantification was established.

*Gallic acid (compound***2***)*: 1 mg of the fraction containing the gallic acid was weighed, dissolved in 1.0 mL of H_2_O:MeOH 55:45, and filtered through a micro-membrane. An aliquot (271 PL) of this solution was separated with the aid of a micropipette and transferred to a volumetric flask (5.0 mL). The flask volume was made up of 55:45 H_2_O:MeOH, providing the stock solution (3.2 × 10^−4^ mM). Starting from the stock solution, six dilutions were prepared, resulting in working solutions with seven different concentrations (5.42 × 10^−2^, 4.64 × 10^−2^, 3.88 × 10^−2^, 3.09 × 10^−2^, 2.33 × 10^−2^, 1.55 × 10^−2^, and 7.75 × 10^−3^).

*Methyl gallate (compound***6***)*: 10.2 mg of the fraction containing the methyl gallate was weighed, dissolved in 5.0 mL of H_2_O:MeOH 55:45 and filtered through a micro-membrane, providing the stock solution (1.1 × 10^−2^ mM). Starting from the stock solution, six dilutions were prepared, resulting in working solutions with seven different concentrations (2.05 × 10^−1^, 1.76 × 10^−1^, 1.46 × 10^−1^, 1.17 × 10^−1^, 8.78 × 10^−2^, 5.85 × 10^−2^, and 2.92 × 10^−2^).

*Myricetin-3-O-rhamnopyranoside (compound***3***)*: One mg of the fraction containing compound 3 was weighed, dissolved in 1 mL of H_2_O:MeOH 55:45, and filtered through a micro-membrane. An aliquot (792 PL) of this solution was separated with the aid of a micropipette and glued in a volumetric flask (5.0 mL). The flask volume was made up of 55:45 H_2_O:MeOH, providing the stock solution (1.8 × 10^−4^ mM). Starting from the stock solution, six dilutions were prepared, resulting in working solutions with seven different concentrations (1.58 × 10^−1^, 1.35 × 10^−1^, 1.13 × 10^−1^, 9.06 × 10^−2^, 6.78 × 10^−2^, 4.52 × 10^−2^, and 2.26 × 10^−2^).

*Myricetin-3-O-(2″-O-galloyl)-α-rhamnopyranoside (compound***7***)*: Three mg of the fraction containing the compound 4 was weighed, dissolved in 3 mL of H_2_O:MeOH 55:45, and filtered through a micro-membrane. An aliquot (2.5 mL) of this solution was separated with the aid of a micropipette and placed in a volumetric flask (5 mL). The flask volume was made up of 55:45 H_2_O:MeOH, providing the stock solution (4.2 × 10^−4^ mM). Starting from the stock solution, six dilutions were prepared, resulting in working solutions with seven different concentrations (5.00 × 10^−1^, 4.30 × 10^−1^, 3.59 × 10^−1^, 2.86 × 10^−1^, 2.13 × 10^−1^, 1.43 × 10^−1^, and 7.17 × 10^−2^).

The calibration method used was the external standard method. The seven standards and the EtOAc fraction were analyzed in a chromatogram obtained via analytical HPLC-PDA in triplicate, using “Luna” Phenomenex LC-18 analytical column, in gradient mode H_2_O/MeOH 95:5 to H_2_O/MeOH 55:45 (20 min), followed by an isocratic mode of H_2_O/MeOH 55:45 for 30 min, with a flow of 1 mL/min, detection at 254 nm, and an injection volume of 20 µL. Then, the response curves were constructed with the mean values of the area relationships of the substances on the ordinate axis and the respective concentration values on the abscissa axis. From the analytical parameters obtained from the calibration curves, it was possible to calculate the limits of detection (LOD) and quantification (LOQ) by the IUPAC method (Table 1). The analytical curve of substances 1, 2, 3, and 4 showed good linearity in the concentration range used. The value of the correlation coefficient obtained, (R), was above 0.991, in accordance with the minimum criterion acceptable by ANVISA (which must be greater than 0.990).

Gallic acid quantification showed an analytical curve with good linearity in the concentration range used (0.01 to 0.06 mg mL^−1^); the value of the correlation coefficient obtained, (R)2, was 0.992, in accordance with the minimum criterion acceptable by the regulatory agency (ANVISA) (which must be greater than 0.990). From the analytical parameters obtained from the calibration curves, it was possible to calculate the limits of detection (LOD) and quantification (LOQ) by the IUPAC method (Table 3). The sensitivity of the method obtained was 9.6 × 10^5^, which corresponds to the slope of the line (slope of the curve) [33]. The UV absorption area of gallic acid present in the ethyl acetate fraction was 141,783.0. Regarding the methyl gallate, good linearity was also observed in the concentration range used (0.03 to 0.21 mg/mL); the value of the correlation coefficient obtained, (R)2, was 0.994. The limits of detection (LOD) and quantification (LOQ) were calculated by the IUPAC method from the analytical parameters obtained from the calibration curves (Table 3). The sensitivity of the method obtained was 1 × 10^5^, which corresponds to the slope of the line (slope of the curve) [33]; the UV absorption area of the methyl gallate present in the ethyl acetate fraction was 112,473.63. For the flavonoid myricetin-3-O-rhamnoside, good linearity was obtained in the concentration range used (0.02 to 0.16 mg/mL); the value of the correlation coefficient obtained, (R)^2^, was 0.994. The sensitivity of the method obtained was 5.2 × 10^6^, which corresponds to the slope of the line (slope of the curve) [33]. The UV absorption area of this compound was 878,408.2. For the flavonoid myricetin-3-O-(2″-*O*-galloyl)-α-rhamnopyranoside, good linearity was also obtained in the concentration range that was used (0.1 to 0.6 mg/mL), and the value of the coefficient of correlation (R)^2^ was 0.991. The sensitivity of the method obtained was 3.4 × 10^6^, which corresponds to the slope of the line (slope of the curve) [33]. The UV absorption area of the flavonoid in this experiment was 629,357.0.

## 4. Conclusions

*Inga* species are promising sources of therapeutic antitumor agents, due mainly to their polyphenols, such as phenolic acids and flavonoids. These metabolites are widely known in the literature to have antitumor potential, and they were found at high concentrations in the fractions with cytotoxic potential. Because these compounds showed a protective effect against damage induced by hydrogen peroxide, studies in animal models are needed to confirm their antitumor potential. The current investigations regarding the proteomic approach demonstrated different changes in the expression of proteins involved in DNA damage repair following treatment with polyphenols, even though they were chemically and structurally very similar. These findings were corroborated by the antigenotoxicity assays and confirm the protective effect of the flavonoids against H_2_O_2_ damage. In addition, they identified a large number of proteins involved in DNA repair and response to ROS generation. However, it was not possible to trace a specific repair pathway, possibly because the cellular machinery does not always present itself in a specific manner, but is instead multifunctional. Thus, this study contributed to the knowledge of the chemical and pharmacological potential of the *Inga* genus, which is widely used as a remedy by several populations.

## Figures and Tables

**Figure 1 molecules-27-04695-f001:**
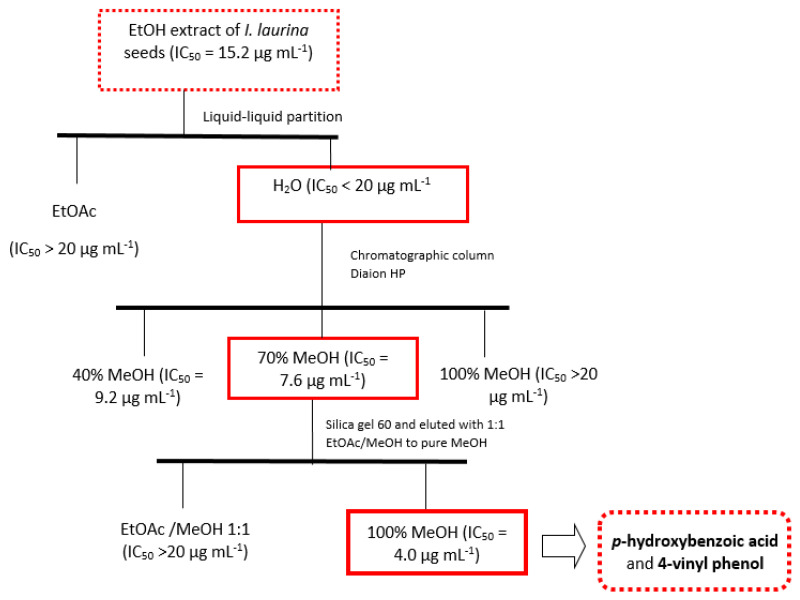
Flow chart showing extraction procedures used to obtain cytotoxic compounds from eth-anolic extract of *I. laurina* seeds.

**Figure 2 molecules-27-04695-f002:**
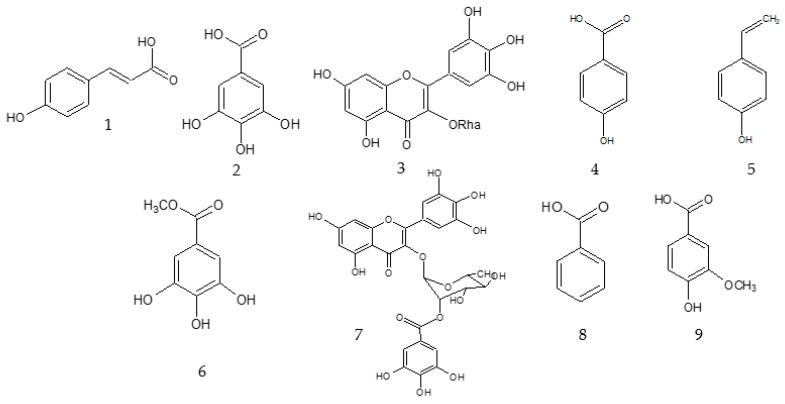
Chemical structure of the compounds (*p*-coumaric acid (**1**), gallic acid (**2**), myricetin-3-*O*-rhamnopyranoside (**3**), *p*-hydroxybenzoic acid (**4**), 4-vinyl phenol (**5**), methyl gallate (**6**), myricetin-3-O-(2″-*O*-galloyl)-α-rhamnopyranoside (**7**), benzoic acid (**8**), and vanillic acid (**9**)) identified in the cytotoxic *Inga* species.

**Table 1 molecules-27-04695-t001:** Inhibitory concentration (IC_50_; µg mL^−1^) values were obtained from the cytotoxic effect of the ethanolic extracts of the three *Inga* species against HT-29 cancer cell lines.

Extracts	*I. laurina*	*I. edulis*	*I. marginata*
Branches	>20	>20	13.8
Flowers	>20	-	-
Fruit peel	>20	-	
Fruit pulp	>20	-	-
Seeds	15.2	-	-
Doxorubicin	1.25		

**Table 2 molecules-27-04695-t002:** Inhibitory concentration (IC_50_; µg mL^−1^) values were obtained from the cytotoxic effect of the ethanolic extracts of the three Inga species against T98G human glioblastoma cancer cell lines.

Extracts	*I. laurina*	*I. edulis*	*I. marginata*
Branches	>20	>20	>20
Flowers	>20	>20	-
Fruit pulp	>20	>20	-
Fruit peel	>20	18.6	-
Seeds	14.4	>20	-
Doxorubicin	2.10		

**Table 3 molecules-27-04695-t003:** Calibration data of the compounds (gallic acid (**2**), methyl gallate (**6**), myricetin-3-O-rhamnoside (**3**), and myricetin-3-O-(2″-*O*-galloyl)-α-rhamnopyranoside (**7**)) to calculate the limits of detection (LOD) and quantification (LOQ) by the IUPAC method [33].

Compound	Line Equation y = ax + b	R^2^	LOD	LOQ	DP
**2**	y = 963,313.28x + 14,384.16	0.992	3.27 × 10^−3^	1.09 × 10^−2^	4500.44
**6**	y = 999,970.55x + 20,108.94	0.995	0.045	0.15	14,533.01
**3**	y = 523,4447.0x + 18,893.8	0.994	0.01	0.04	19,471.80
**7**	y = 3,410,900.0x + 10,0491.6	0.991	0.09	0.30	10,1824.1

**Table 4 molecules-27-04695-t004:** Identification of proteins detected from HepG2 cells treated with flavonoid myricetin-3-rhamnoside and myricetin-3-O-(2″-*O*-galloyl)-α rhamnopyranoside with a score ≥ 30.

Myricetin-3-Rhamnoside			
Prot Acc	Protein	Score	116/114
K6PP_HUMAN	6-phosphofructokinase type C OS = Homo sapiens GN = PFKP PE = 1 SV = 2	208	0.00
CH60_HUMAN	60 kDa heat shock protein, mitochondrial OS = Homo sapiens GN = HSPD1 PE = 1 SV = 2	141	0.00
G6PI_HUMAN	Glucose-6-phosphate isomerase OS = Homo sapiens GN = GPI PE = 1 SV = 4	82	0.00
CYB5B_HUMAN	Cytochrome b5 type B OS = Homo sapiens GN = CYB5B PE = 1 SV = 2	44	0.00
QCR6_HUMAN	Cytochrome b-c1 complex subunit 6, mitochondrial OS = Homo sapiens GN = UQCRH PE = 1 SV = 2	38	0.00
ST1A3_HUMAN	Sulfotransferase 1A3/1A4 OS = Homo sapiens GN = SULT1A3 PE = 1 SV = 1	37	0.00
NQO1_HUMAN	NAD(P)H dehydrogenase [quinone] 1 OS = Homo sapiens GN = NQO1 PE = 1 SV = 1	45	0.20
ALDR_HUMAN	Aldose reductase OS = Homo sapiens GN = AKR 1B1 PE = 1 SV = 3	65	0.22
ERO1A_HUMAN	ERO1-like protein alpha OS = Homo sapiens GN = ERO1L PE = 1 SV = 2	36	0.27
CHSP1_HUMAN	Calcium-regulated heat stable protein 1 OS = Homo sapiens GN = CARHSP1 PE = 1 SV = 2	60	0.28
ANXA2_HUMAN	Annexin A2 OS = Homo sapiens GN = ANXA2 PE = 1 SV = 2	185	0.45
TERA_HUMAN	Transitational endoplasmic reticulum ATPase OS = Homo sapiens GN = VCP PE = 1 SV = 4	86	0.48
FLNA_HUMAN	Filamin-A OS = Homo sapiens GN = FLNA PE = 1 SV = 1	48	0.49
PROF1_HUMAN	Profilin-1 OS = Homo sapiens GN = PFN1 PE = 1 SV = 2	48	0.49
PSA7_HUMAN	Proteasome subunit alpha type-7 OS = Homo sapiens GN = PSMA7 PE = 1 SV = 1	54	1.66
VIME_HUMAN	Vimentin OS = Homo sapiens GN = VIM PE = 1 SV = 4	278	1.68
GFAP_HUMAN	Glial fibrillary acidic protein OS = Homo sapiens GN = GFAP PE = 1 SV = 1	46	1.71
PDIA1_HUMAN	Protein disulfide-isomerase OS = Homo sapiens GN = P4HB PE = 1 SV = 3	42	1.79
myricetin-3-O-(2″-*O*-galloyl)-α rhamnopyranoside			
Prot Acc	Protein	Score	117/114
TPIS_HUMAN	Triosephosphate isomerase OS = Homo sapiens GN = TPI1 PE = 1 SV = 3	105	0.00
G6PI_HUMAN	Glucose-6-phosphate isomerase OS = Homo sapiens GN = GPI PE = 1 SV = 4	82	0.00
TYPH_HUMAN	Thymidine phos’phorylase OS = Homo sapiens GN = TYMP PE = 1 SV = 2	59	0.00
PSA7_HUMAN	Proteasome subunit alpha type-7 OS = Homo sapiens GN = PSMA7 PE = 1 SV = 1	54	0.00
H4_HUMAN	Histone H4 OS = Homo sapiens GN = HIST1H4A PE = 1 SV = 2	36	0.00
FLNA_HUMAN	Filamin-A OS = Homo sapiens GN = FLNA PE = 1 SV = 4	48	0.09
PROF1_HUMAN	Profilin-1 OS = Homo sapiens GN = PFN1 PE = 1 SV = 2	48	0.20
MVP_HUMAN	Major vault protein OS = Homo sapiens GN = MVP PE = 1 SV = 4	80	0.36
CHSP1_HUMAN	Calcium-regulated heat stable protein 1 OS = Homo sapiens GN = CARHSP1 PE = 1 SV = 2	60	0.39
PYGB_HUMAN	Glycogen phosphorylase, brain form OS = Homo sapiens GN = PYGB PE = 1 SV = 5	39	0.43
TBB3_HUMAN	Tubulin beta-3 chain OS = Homo sapiens GN = TUBB3 PE = 1 SV = 2	160	0.45
BLVRB_HUMAN	Flavin reductase (NADPH) OS = Homo sapiens GN = BLVRB PE = 1 SV = 3	85	0.50
CYB5B_HUMAN	Cytochrome b5 type B OS = Homo sapiens GN = CYB5B PE = 1 SV = 2	44	1.65
ERO1A_HUMAN	ERO1-like protein alpha OS = Homo sapiens GN = ERO1L PE = 1 SV = 2	36	1.72
MYOF_HUMAN	Myoferlin OS = Homo sapiens GN = MYOF PE = 1 SV = 1	61	1.86

## Data Availability

Not applicable.

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
