# Peer review of "Metabolic Profiling of Inga Species with Antitumor Activity"

_molecules, 2022, doi:10.3390/molecules27154695_

Round 1
Reviewer 1 Report
This manuscript describes bioassay-guided fractionation of the extracts from Inga plants and identification of some phenolic compounds in the bioactive fractions. The bioassay was based on the determination of the viability of HT-29 and T98G cancer cells. Proteomics analysis of HepG2 cancer cells treated with antigenotoxic concentrations of myricetin derivatives found in the fractions was then performed. The described findings are interesting and may have potential clinical applicability. However, there are several serious concerns about the data presentation, description and interpretation, as follows.
Major points:
1. The numbers presented in lines 228 and 231 (eg 18.48 ± 27.61, 10.84 228± 14.43 and 12.23 ± 17.9) might be DNA percentages but this is unclear since the description of the antigenotoxicity results is incomprehensive.
Also, the above numbers are meaningless unless the data from the negative control (untreated) cells and the positive control cells treated with H2O2 alone are provided. These results must be illustrated, eg on a graph, along with examples of the original images of the slides subjected to the comet assay, including both the negative and positive controls.
Further, in the Methods (“Antigenotoxicity assessment”, line 412), the authors mentioned that “Pre-treatment and post-treatment protocols were used, as proposed by Scolastici et al (2008)”. How are these protocols reflected in the Results? In addition, there are many repetitions in the description of this method.
What is the reason why antigenotoxicity was examined in HepG2 cells and not in HT29 or T98G cells that were used in the cytotoxicity assays?
2. line 604: “The metabolic content analysis highlighted the compounds responsible for cytotoxic activity against the human colon cancer cells” – The data showing the cytotoxic activity of the isolated compounds are not presented. No evidence is provided that these compounds are “responsible for cytotoxic activity” (see Conclusions) of the tested Inga extracts or fractions. Unless such evidence is presented this and similar statements need to be omitted from the narrative.
3. What is the reason why the two flavonoids were selected for the proteomics analysis in HepG2 cells?
4. The Discussion is scarce. The roles of the differentially expressed proteins in the antigenotoxic activity of myricetin derivatives are not well summarized. The authors need to describe the potential efficacy of the identified compounds when compared to current chemotherapies. Please elaborate and emphasize the potential clinical applicability and the novelty of the findings.
5. The Conclusions contain reiteration of the results, which doesn’t seem to be appropriate (lines 607-611).
6. line 386: “4.7.1. Cytotoxicity assay”. The assay is described only for HepG2 cells. It is unclear how the viability of the two other cell lines was determined. The description provided in lines 407-411 is unsuitable for the “Methods”.
Technical points.
1. A major problem of the manuscript is that it is written in poor technical and scientific English. The narrative (especially in the Results, Discussion, and Conclusions) is not well arranged and is often confusing, with many repetitive phrases, numerous grammar and style errors, erroneous wordings and wrongly constructed phrases. Below are just a few examples:
a) The Abstract does not provide sufficient and consistent information on the main findings the study and should be rewritten. The description of the proteomics results is poor and incomprehensive while the content is mainly related to the cytotoxic effects of the extracts and some of the fractions.
“This work evaluated the proteomic and metabolomic profiles of Inga species. The experimental procedures used in this study were based on the comet assay and were verified by detailed proteomic analysis of the HepG2 cells following treatment with the isolated bioactive compounds.” – Not all the procedures “used in this study” were based on the comet assay. Did the authors perform proteomics experiments in order to verify the procedures?
“We found proteins directly related to the elimination of ROS, DNA repair, expression of tumor proteins, and apoptosis.” – These proteins have long been “found”. Also, the following expression is meaningless: “We found proteins directly related to … expression of tumor proteins”.
b) line 77: "... the branches extract of I. marginata (IC50 =13.8 ug mL-1) and I. laurina seeds (IC50 = 15.2 ug mL-1)."
c) line 83: "… antitumor activity (IC50 = 13.8 ug mL-1) in the evaluated assay" – This sounds like the authors evaluated the assay itself.
d) line 178: "Several studies have reported natural phenolic compounds as potent bioactive with potential ..."
e) line 209 and elsewhere: "The sensitivity of the method obtained were..."
f) line 223: “The antigenotoxicity and proteomic approach were performed…”
g) line 276: “…a mutation in the protein gene…”
h) line 421: "In order to evaluate the antigenotoxicity of flavonoids from I. laurina, it was used hydrogen peroxide..."
j) line 436, etc.: "Each blade was covered..." – Did the authors use blades?
k) line 512: "It weighed 7 x 10-1 mg of the EtOAc fraction in an analytical balance"
l) line 222: “2.3. Antigenotoxicity and Proteomic approach from the flavonoids”
2. The title of the manuscript is overambitious and incorrect. While phenolic metabolites were profiled in some fractions of the Inga plant extracts, the proteomic analysis was done in human cells and not in “cytotoxic Inga species”. Also, only two out of multiple compounds from the Inga species were used in the proteomic assays. Thus, the title and similar sentences in the Abstract and elsewhere need to be revised.
3. The Abstract mainly contains the description of the cytotoxicity data
4. The names of natural phenolic compounds, such as gallic acid, should not be capitalized. These are not trade names.
5. line 125: “Therefore, all extracts of plant material from Inga species were submitted to cytotoxicity analysis against T98G human glioblastoma cancer cell lines, using doxorubicin as positive control (IC50 = 2.10 μg mL-1). Table 2 shows that except for the extract of I. laurina seeds and I. edulis fruit peel, the plant material obtained from the three species of Inga showed no cytotoxic effect against this tumor cell line.”
- The meaning of the cytotoxicity data obtained in T98G cells is not explained. The description of these results doesn’t seem to be related with the previous paragraphs, so why “Therefore”? Both this description and Table 2 might be misplaced because both the preceding and the following content address the fractionation results based on the cytotoxic effect on HT-29 cells.
6. “To determine the protein profile, cells treated and untreated with the two flavonoids of I. laurina were used.” - What cells and what flavonoids?
7. In Figure 2, numbers 5 and 6 are missing.
8. Some of the references are not formatted, e.g. Scolastici et al (2008), etc.
In summary, the manuscript needs to be rewritten and proofread by an English-speaking person familiar with this kind of research.
Author Response
Dear Reviewer,
Attached, you will find the point-by-point response to the reviewer’s comments.
With best regards

Reviewer 2 Report
Current report evaluated the proteomic and metabolomic profiles of Inga species. Please conduct the concerns below.
1. The title seems better to revise from “Cytotoxic Inga Species” as “Antitumor Inga Species”.
2. How many Inga plants were employed to evaluate the antigenotoxicity? It must show in the abstract in clear.
3. Extraction procedures must follow the established method with reference(s).
4. In Table 3, the IUPAC method must indicate the reference in legends.
5. Proteomic Analysis needs the reference(s) to support.
6. Cytotoxic effects of phenolic acids including Gallic acid and Methyl gallate or flavonoids such as Myricetin-3-O-rhamnoside and Myricetin-3-O-(2"-O-galloyl)-α-rhamnopyranoside must follow the previous report(s).
7. New compound(s) and novelty of finding(s) must describe in conclusion in clear.
Author Response

(The authors gave the same response as above.)

Reviewer 3 Report
The introduction is feeble. The authors describe a reference as a paragraph. They must improve the introduction.
The methodology is deficient. The proteomics is based on Maldi-Tof; however, the authors talked about the differential expression of proteins. How did they determine the differential expression of proteins? Also, they did not describe how they sequenced the peptides. They needed to digest the proteins via trypsin (as an example) to obtain the peptides and sequence by Malfi-Tof or LC-MS/MS.
The tables where the authors described the results did not show the coverage of peptides or peptide sequence. Please indicate the coverage peptides parameter and sequences of peptides.
The discussion and conclusion are feeble.
The bibliography is not adequate. The authors need to refresh the bibliography; please use the reference 2017-2022.
Author Response

(The authors gave the same response as above.)

Round 2
Reviewer 1 Report
The revised manuscript is somewhat improved. However, several comments were not fully addressed or were ignored:
1. … These results must be illustrated, eg on a graph, along with examples of the original images of the slides subjected to the comet assay, including both the negative and positive controls.
Answer: The authors agree with all points raised, which have been incorporated in this new version. The manuscript has been modified as suggested. The numbers presented (eg 18.48 ± 27.61, 10.84 228± 14.43 and 12.23 ± 17.9) describe be DNA percentages and standard error (% DNA ± SE. The results of both the negative and positive controls were added to the text (% DNA ± SE of the negative control is 10.71 ± 1.05*** and the positive control is 36.21 ± 3.33).
- It is hard to evaluate these results based on the numbers presented in the text. “the results MUST BE ILLUSTRATED, EG ON A GRAPH, ALONG WITH EXAMPLES OF THE ORIGINAL IMAGES OF THE SLIDES SUBJECTED TO THE COMET ASSAY”.
… How are these protocols reflected in the Results?
Answer: Thank you for the contribution, see correction suggested in the revised manuscript. The method has been rewritten in order to avoid repetitions. Concerning the reference mentioned above, the protocols proposed by Scolastici et al. (2008) are widely used to assess the antigenotoxicity of natural products and identify metabolites with antigenotoxic and chemopreventive action. The protocols employed in this study provided satisfactory results.
- The authors should clearly indicate and illustrate in the Results the data obtained with the “pre-treatment protocol” and those obtained with the “post-treatment protocol”.
Technical points.
Several technical comments have been ignored by the authors and their answers are misleading:
b) line 77: "... the branches extract of I. marginata (IC50 =13.8 ug mL-1) and I. laurina seeds (IC50 = 15.2 ug mL-1)."
Answer: Correction done.
- NOT DONE
d) line 178: "Several studies have reported natural phenolic compounds as potent bioactive with potential ..."
Answer: Correction done.
- NOT DONE
e) line 209 and elsewhere: "The sensitivity of the method obtained were..."
Answer: Correction done.
- NOT DONE. In addition to the grammar errors, the meaning of the numbers in this sentence is unclear. What are the measurement units?
g) line 276: “…a mutation in the protein gene…”
Answer: Correction done.
- NOT DONE
j) line 436: "Each blade was covered..." – Did the authors use blades?
Answer: Correction done.
- NOT DONE
k) line 512: "It weighed 7 x 10-1 mg of the EtOAc fraction in an analytical balance"
Answer: Correction done.
- NOT DONE
l) line 222: “2.3. Antigenotoxicity and Proteomic approach from the flavonoids”
Answer: Correction done.
- NOT DONE
New technical comments:
There are still many previous as well as new errors in the text BEYOND the examples listed below:
1). Divisão de Química Medicinal e Farmacognosia, Faculdade de Farmácia, The Ohio State University, 17Columbus 43210, United States
- This is a US university, so the department's name should be written in English.
2). Lines 87, 132, etc.: “against cancer cell LINES HT-29”
- Only one cell line is mentioned. Also, the word order is wrong.
3). Line 270: “…was regulated by both the treatments”
4). Line 399: “The cells were cultured in DMEM culture medium (Modified Eagle Medium Dulbecco's – Sigma®) supplemented with penicillin 100 U mL-1 and streptomycin 0.1 mg mL-1 sodium bicarbonate, kanamycin (Sigma®), pH 7.2 and supplemented with 10% fetal bovine serum (Cultilab®).
- Incorrect presentation of company names and repetitive words.
Etc.
When responding to comments, please indicate the exact locations (line ##) of the changes made in the text.
Reviewer 2 Report
It has been revised in a good way.
Reviewer 3 Report
The paragraph is very short and seems to be one idea by one paragraph. The authors need to group ideas and form a better paragraph.
Though the authors better the methodology, they don’t describe how to prepare the peptides for analysis by MALDI. Moreover, they tell a gene expression, but how do they explain the gene expression with a qualitative methodology? Probably they wanted to notice that different kinds of proteins were expressed. However, they mentioned that “x” protein was expressed several times. Did they use a densitometer? If they want to report the protein expression, they should perform a western blot or qPCR. Did they trypsin digestion? If they did not perform trypsin digestion, is a new methodology or methodology modification? The authors need to be clear in this section of methodology.